# Combined Therapy with a CCR2/CCR5 Antagonist and FGF21 Analogue Synergizes in Ameliorating Steatohepatitis and Fibrosis

**DOI:** 10.3390/ijms23126696

**Published:** 2022-06-15

**Authors:** Tobias Puengel, Sander Lefere, Jana Hundertmark, Marlene Kohlhepp, Christian Penners, Frederique Van de Velde, Bruno Lapauw, Anne Hoorens, Lindsey Devisscher, Anja Geerts, Stephanie Boehm, Qihong Zhao, John Krupinski, Edgar D. Charles, Bradley Zinker, Frank Tacke

**Affiliations:** 1Department of Hepatology & Gastroenterology, Charité—Universitätsmedizin Berlin, Campus Virchow-Klinikum (CVK) and Campus Charité Mitte (CCM), 13353 Berlin, Germany; jana.hundertmark@charite.de (J.H.); marlene.kohlhepp@charite.de (M.K.); frank.tacke@charite.de (F.T.); 2Berlin Institute of Health (BIH), 10178 Berlin, Germany; 3Department of Medicine III, RWTH-University Hospital Aachen, 52074 Aachen, Germany; cpenners@ukaachen.de; 4Hepatology Research Unit, Department of Internal Medicine and Pediatrics, Liver Research Center Ghent, Ghent University, B-9000 Ghent, Belgium; anja.geerts@uzgent.be; 5Department of Endocrinology, Ghent University, B-9000 Ghent, Belgium; frederique.vandevelde@ugent.be (F.V.d.V.); bruno.lapauw@ugent.be (B.L.); 6Department of Pathology, Ghent University Hospital, B-9000 Ghent, Belgium; anne.hoorens@uzgent.be; 7Gut-Liver Immunopharmacology Unit, Department of Basic and Applied Medical Sciences, Liver Research Center Ghent, Ghent University, B-9000 Ghent, Belgium; lindsey.devisscher@ugent.be; 8Bristol-Myers Squibb, Princeton, NJ 08540, USA; stephanie.boehm@bms.com (S.B.); qihong.zhao@bms.com (Q.Z.); john.krupinski@bms.com (J.K.); edgar.charles@bms.com (E.D.C.); bradley.zinker@bms.com (B.Z.)

**Keywords:** macrophages, monocytes, treatment strategies, chemokines, inflammation, fibroblast growth factor (FGF), nonalcoholic steatohepatitis (NASH), fibrosis, nonalcoholic fatty liver disease (NAFLD), metabolism

## Abstract

(1) Background: With new potential drug targets emerging, combination therapies appear attractive to treat non-alcoholic steatohepatitis (NASH) and fibrosis. Chemokine receptor CCR2/5 antagonists can improve fibrosis by reducing monocyte infiltration and altering hepatic macrophage subsets. Fibroblast growth factor 21 (FGF21) may improve NASH by modulating lipid and glucose metabolism. We compared effects of single drug to combination treatment as therapeutic strategies against NASH. (2) Methods: We analyzed serum samples and liver biopsies from 85 nonalcoholic fatty liver disease (NAFLD) patients. A CCR2/5 inhibitor (BMS-687681-02-020) and a pegylated FGF21 agonist (BMS-986171) were tested in male C57BL/6J mice subjected to dietary models of NASH and fibrosis (choline-deficient, L-amino acid-defined, high-fat diet (CDAHFD) up to 12 weeks; short- (2w) or long-term (6w) treatment). (3) Results: In NAFLD patients, chemokine CCL2 and FGF21 serum levels correlated with inflammatory serum markers, only CCL2 was significantly associated with advanced liver fibrosis. In rodent NASH, CCR2/5 inhibition significantly reduced circulating Ly6C+ monocytes and hepatic monocyte-derived macrophages, alongside reduced hepatic inflammation and fibrosis. FGF21 agonism decreased body weight, liver triglycerides and histological NASH activity. Combination treatment reflected aspects of both compounds upon short- and long-term application, thereby amplifying beneficial effects on all aspects of steatohepatitis and fibrosis. (4) Conclusions: CCR2/5 inhibition blocks hepatic infiltration of inflammatory monocytes, FGF21 agonism improves obesity-related metabolic disorders. Combined therapy ameliorates steatohepatitis and fibrosis more potently than single drug treatment in rodent NASH, corroborating the therapeutic potential of combining these two approaches in NASH patients.

## 1. Introduction

Non-alcoholic fatty liver disease (NAFLD) is still increasing in prevalence and is the most common liver disease worldwide [1]. Metabolic syndrome caused by “Western lifestyle” with its key characteristics obesity, lipometabolic disorder, impaired glucose tolerance, insulin resistance and type 2 diabetes is growing alongside, constituting a major risk for the development of NAFLD [2]. The term NAFLD covers different entities such as non-alcoholic fatty liver (NAFL) and its inflammatory form non-alcoholic steatohepatitis (NASH). Chronic inflammatory processes result in liver fibrosis and ultimately liver cirrhosis, which is associated with a risk for developing hepatocellular carcinoma (HCC) [3]. Liver fibrosis appears to be the dominant predictor of disease-specific mortality in NAFLD [4,5,6,7]. Over the last years, translational and clinical studies have identified many potential drug targets including metabolic, inflammatory and cell death pathways [2,8,9]. Nevertheless, at present, available data indicate a limited efficacy of single compounds in treating patient-relevant endpoints in NAFLD [10]. In the interim analysis of a phase III clinical trial, high-dose obeticholic acid was able to improve liver fibrosis without worsening NASH, showing at least a 1-point improvement in key features of liver histology scoring, in only 23% of patients as compared to 12% in placebo controls [11]. Combining drugs with different targets could be a promising strategy to increase efficacy of NAFLD treatment, particularly for “difficult-to-reach” endpoints such as fibrosis regression [2,9].

Among the potential targets in NAFLD, myeloid liver cells display a functionally distinct, inflammatory phenotype [12,13,14]. In mice, pharmacologically targeting the chemokine (C-C motif) ligand 2 (CCL2)—chemokine receptor (C-C motif) 2 (CCR2) pathway reduces monocyte infiltration and accumulation of monocyte-derived macrophages (MoMF) into the injured liver [15,16]. Modulation of the hepatic macrophage pool finally results in amelioration of liver fibrosis and steatohepatitis in mouse models [17]. In a phase II clinical trial in NASH patients (*n* = 289), the dual CCR2/CCR5 inhibitor cenicriviroc has demonstrated anti-fibrotic efficacy after one year of treatment [18]. However, the treatment benefit did not remain significant over two years of cenicriviroc therapy [19], and the further development of the drug in this indication has been terminated due to lack of clear efficacy signals of cenicriviroc monotherapy upon interim analysis of a phase III clinical trial [20].

Among the potential metabolic targets in NAFLD, fibroblast growth factor 21 (FGF21) is a systemically circulating and liver-derived hormone achieving organ specificity by the tissue distribution of the FGF receptor and its co-receptor β-Klotho, which is mainly expressed in hepatic and adipose tissue [21,22]. In contrast to other members of the FGF superfamily, FGF21 acts in both a paracrine and endocrine manner, as it does not bind to heparan sulfate [23]. FGF21 displays multiple metabolic effects. As such, FGF21 agonism has been shown to reverse hepatic fat infiltration, with additional effects on insulin sensitivity, mediated by increased glucose uptake in adipocytes [24]. In mice, FGF21 administration has been shown to increase fatty acid oxidation and lower blood glucose as well as triglyceride levels protecting animals from diet induced obesity and diabetes [25,26]. The PEGylated human analogue pegbelfermin (BMS-986036) reduced hepatic fat content and improved other surrogate measures of NASH and metabolic markers in a phase II trial in NASH patients [27]. The FALCON program investigated efficacy and safety of pegbelfermin in patients with NASH and stage 3 fibrosis (FALCON1, ClinicalTrials.gov Identifier NCT03486899) as well as compensated NASH cirrhosis (FALCON2, ClinicalTrials.gov Identifier NCT03486912). Although primary endpoints were not met for either study, in stage 3 fibrosis improvements in biopsy-assessed fibrosis and NAFLD activity score (NAS) were observed. Improvements in non-invasive surrogate markers (liver fat, inflammation and fibrosis) were observed upon pegbelfermin treatment in both studies [28,29,30].

In the pathogenesis of steatohepatitis and liver fibrosis, not one single pathway is dysregulated, but multiple events combine in the disease pattern of NAFLD [31,32,33]. Therefore, as new potential drug targets emerge, combination therapies could possibly bring advantages over single drug treatments and/or sustain therapeutic benefits [32]. A particularly promising option appears to be the combination of antifibrotic and metabolic drugs [34].

In this study, we explored the therapeutic efficacy of CCR2/5 inhibition and FGF21 agonism using two novel pharmacological compounds in mouse models of liver injury and NASH. Our data suggest beneficial and even additive effects by combining these drug treatments, mandating clinical studies of combination therapies in patients with NASH and liver fibrosis.

## 2. Results

### 2.1. CCL2 and FGF21 Serum Levels Correlate with Different Aspects of Human NAFLD

In order to establish the relevance of targeting CCR2/CCL2 and FGF21 in humans, we analyzed serum CCL2 and FGF-21 levels in 85 patients with biopsy-confirmed NAFLD. CCL2, also called monocyte chemoattractant protein-1 (MCP-1), is expressed and secreted by various hepatic cells during fibrosis progression, as shown in mouse models as well as human patients [35,36]. FGF21 is synthesized and secreted from the liver and has multiple metabolic effects [24]. Deficient or aberrant FGF21 is associated with NAFLD/NASH, and elevated FGF21 serum levels correlate with hepatic fat content in mice and humans (reviewed in [37]). Patients in our cohort were classified as having non-alcoholic fatty liver (NAFL; *n* = 31) or non-alcoholic steatohepatitis (NASH; *n* = 54), based on liver biopsies evaluated by an expert pathologist (Appendix A) and supported by clinical and biochemical patient characteristics (Table 1). Fibrosis was absent, mild or moderate (F0-F2) in 73 patients, while 12 patients had progressed to advanced fibrosis (F3-F4). CCL2 serum levels were significantly elevated in NAFLD patients with advanced fibrosis compared to those without (*p* < 0.001), and also correlated with advanced fibrosis (F3-4 fibrosis based on histopathology and FIB-4 score) (Figure 1A,C; Appendix A). On the other hand, CCL2 levels were not significantly associated with NASH activity (NAS, GGT, AST) (Figure 1A,C; Appendix A). These associations were principally found in male and female patients but did not reach significance in females due to the lower number of advanced disease stages in females in our cohort (Appendix A). FGF21 serum levels did not correlate with the stage of fibrosis in the whole cohort, nor did CCL2 concentrations correlate with FGF21 serum levels (Figure 1B,D; Appendix A). However, FGF21 was associated with biomarkers of steatohepatitis (CK-18 fragment M30, GGT, AST) (Figure 1E). In multivariate analysis, the association between advanced fibrosis (F3-F4) and CCL2 levels remained significant (*p* = 0.017) after adjusting for the AST/ALT ratio and presence of type 2 diabetes, which were the only other factors independently associated with advanced fibrosis in our cohort (Appendix A). In conclusion, these data indicate differing roles for CCR2-CCL2 and FGF21 pathways in the pathogenesis of liver fibrosis and chronic steatohepatitis in human NAFLD, corroborating the potential distinct benefits of pharmacological targeting each pathway.

### 2.2. Combination Therapy by Dual CCR2/CCR5 Inhibition and FGF21 Agonism Ameliorates Steatohepatitis and Fibrosis More Effectively Than Single Drug Treatment

In mice and humans, infiltration of monocytes and accumulation of MoMF into the chronically injured liver can be inhibited by CCR2/5 antagonists [15,17], while fibroblast growth factor 21 (FGF21) can ameliorate pathogenic drivers of NASH and fibrosis by affecting lipid and glucose metabolism [22,26]. To evaluate therapeutically targeting those pathways we employed the choline-deficient, L-amino acid-defined, high-fat diet (CDAHFD) liver injury model to induce steatohepatitis and liver fibrosis over a total period of 12 weeks and started pharmacological treatment at week 7 over the last 6 weeks (Figure 2A). Target engagement for efficient pharmacologic inhibition of CCR2 and CCR5 was confirmed by elevated levels of CCL2 and CCL5 (i.e., the respective ligands) in the serum (Appendix A). Both, vehicle and compound treatment were very well tolerated—no mouse had to be ruled out from the study groups. Control animals showed a continuous weight gain over time, while we observed a model specific, initial weight loss followed by stabilization of the bodyweight to baseline levels in CDAHFD fed mice. In comparison to vehicle groups, mice that received PEG-FGF21v demonstrated moderate weight loss, as anticipated, unlike CCR2/5 inhibitor-treated mice. Combination treatment showed additive effects, as mice displayed the lowest bodyweight overall (Figure 2B). Inhibition of CCR2/CCR5 was accompanied by significantly reduced infiltration of hepatic macrophages (Figure 2C,I), serum alanine transaminase (ALT) levels (Figure 2D) and liver fibrosis (Figure 2C,E). Assessment of the NAFLD activity score (NAS) components revealed beneficial effects of FGF21 administration on hepatic steatosis, which was confirmed by a lower hepatic triglyceride content, while CCR2/5 inhibitor treatment was associated with reduced lobular inflammation (Figure 2B,E,F). Whereas an ALT decrease was observed following PEG-FGF21v treatment, liver fibrosis was mainly mitigated by CCR2/5 inhibition (Figure 2C–E).

Importantly, combination treatment with the CCR2/5 antagonist and the PEG-FGF21v reflected beneficial effects of both single drug treatments regarding body weight evolution, hepatic triglyceride content, histological liver injury and fibrosis. In addition, combined therapy even had additive effects on hepatocyte ballooning and on the NAS overall. Positive additive effects were also revealed on gene expression levels related to inflammation (*Tnf*, *Mcp1*) and fibrosis (*Tgfβ*, *αsma*, *Col1a1*, *Timp1*) (Appendix A). CCR2/5 inhibitor-treated animals demonstrated a strong inhibition of blood monocytes (Figure 2H) and F4/80-positive macrophage accumulation in the liver (Figure 2C,J). Flow cytometric characterization of liver macrophages demonstrated that CCR2/5 inhibition specifically reduced infiltrating MoMF neither affecting resident hepatic macrophages (Figure 2I) nor blood and liver lymphocyte populations (Appendix A). These data suggest that combination of CCR2/5 inhibition and PEG-FGF21v treatment is even more potent than single drug treatment alone.

### 2.3. Effects of CCR2/CCR5 Inhibition and FGF21 Agonism on Hepatic Infiltration of Monocytes in Acute Liver Injury

Surprisingly, CCR2/5 inhibition and PEG-FGF21v treatment resulted in a comparable reduction in hepatic macrophages in experimental NASH, raising the question whether both would act directly on inflammatory cells. The chemokine CCL2 is known to attract monocytes via CCR2 to the site of injury leading to the differentiation of monocytes into MoMF [35,38]. In order to investigate the impact of both compounds on inflammatory cell recruitment, we employed an acute liver injury model induced by a single injection of carbon tetrachloride (CCl_4_) in mice. This model induces a sterile injury with strong cell recruitment [15]. Immune cell populations and liver injury were assessed 36 h after CCl_4_ injection (Figure 3A). As expected, CCR2/5 inhibition was associated with significantly reduced numbers of hepatic monocytes and F4/80-positive hepatic monocyte-derived macrophages (MoMF) (Figure 3B,C,G) as well as blood monocytes (Figure 3F). The reduction in monocytes and MoMF was accompanied by a significant amelioration of the liver injury, as assessed by quantification of the necrotic area fraction (Figure 3B,C) and serum ALT and AST levels (Figure 3D). Of note, CCR2/5 inhibition did not affect other myeloid or lymphoid immune cell populations. In corroboration of these findings, combination therapy reflected the inhibition of monocyte infiltration into acutely injured liver that was also seen by CCR2/5 inhibitor single treatment. Interestingly, PEG-FGF21v treatment was associated with a trend towards reduced levels of blood neutrophils (Figure 3F) and significantly lower AST and ALT serum levels (Figure 3D) without an impact on necrotic area fraction (Figure 3C). In general, these findings support that CCR2/5 inhibition blocks monocyte infiltration in acute liver injury. Importantly, this effect is not affected by combination treatment with PEG-FGF21v, stressing the different mode of action of both compounds.

### 2.4. Potent Additive Effects of Combination Therapy Are Already Active at Early Disease Stages

Based on the positive effects of single and combination therapy on histological endpoints in the 6-weeks treatment model, we next investigated the early effects of pharmacological therapy in the CDAHFD mouse model. In contrast to our long-term model, we assessed liver injury after only two weeks of pharmacological treatment, thus at week 8 after injury induction (Figure 4A). Histologic analysis showed moderate levels of fibrosis and steatohepatitis after 8 weeks CDAHFD (Figure 4B). All aspects of disease phenotype were most effectively improved in the combination treatment (Figure 4B–D). Liver injury as assessed by serum alanine transaminases (ALT) was significantly reduced in mice which received PEG-FGF21v. However, liver injury was ameliorated most significantly when PEG-FGF21v was combined with the CCR2/5 antagonist (Figure 4C). Similarly, liver triglycerides were moderately (non-significantly) reduced with PEG-FGF21v, but significantly lower in the combination treatment (Figure 4C). Single drug treatment caused trends (non-significant) towards reduced levels of fibrosis at this timepoint (Figure 4D), but a stronger fibrosis reduction upon combined therapy with CCR2/5 antagonist and PEG-FGF21v. Similarly, although all therapy regimens significantly reduced the NAS, combination treatment was most effective for all aspects of NAS (Figure 4C).

The particular effects of the CCR2/5 antagonist on the composition of the hepatic immune cell compartment were present in single CCR2/5 inhibitor or combination therapy. Analogous to long-term treatment, monocytes and monocyte-derived macrophages were significantly reduced upon pharmacological treatment without affecting resident liver Kupffer cells or lymphoid immune cells (Figure 4F). Collectively, while individual aspects of distinct drug targeting remained preserved in combination treatments, targeting multiple pharmacological pathways appeared more potent for improving the liver disease phenotype than single drug treatment—providing justification for clinical studies of combination treatments of these mechanisms in patients with NASH and fibrosis.

## 3. Discussion

The prevalence of NAFLD and NASH is increasing, leading to the projection that liver-related morbidity and mortality will dramatically increase within the next decades in many areas of the world [39]. At present, lifestyle modification is the mainstay of therapeutic recommendations, while no specific pharmacological treatment is available for the therapy of NAFLD/NASH [40]. With many drugs targeting multiple pathways in metabolism, inflammation and fibrogenesis under development, it can be expected that several drugs will be approved in the foreseeable future [2,10]. Nevertheless, many compounds, including obeticholic acid, cenicriviroc and pegbelfermin, have only improved the investigational endpoints in a subset of patients exposed to these drugs [11,19,27]. The involvement of many pathophysiological mechanisms and the crosstalk between them in NAFLD could partly explain why targeting a single pathway might be insufficient. Combination treatment is therefore an attractive possibility to overcome these problems, although there is currently little evidence to suggest specific combinations.

In this study, we demonstrate that the combination of targeting inflammatory pathways through inhibiting the CCR2 and CCR5 chemokine receptors and reducing lipid deposition in hepatocytes through a pegylated FGF21 analogue ameliorated all histological features of NASH, including liver fibrosis, in mice. The combination regimen had additive effects compared to the use of the single compounds. These data are furthermore substantiated by an analysis of hepatic immune cells, hepatic fat content, serum CCL2 and FGF21 levels, which reflect the distinct pathophysiological modes of action for both compounds.

The “ideal combination” of anti-NASH drugs remains to be determined. However, based on the central role of macrophages as chief regulators of inflammation-induced insulin resistance, hepatic inflammation and fibrogenesis [14], we reasoned that an “anti-inflammatory” compound would be beneficial in a combination therapy regimen. Research from our group and others has demonstrated the therapeutic potential of blocking the CCL2/CCR2-mediated monocyte infiltration in the liver [17,41]. The CCR2/5 inhibitor cenicriviroc was investigated in the phase 2b CENTAUR trial, in which cenicriviroc significantly increased the proportion of patients achieving fibrosis regression after one year of treatment [18]. In our study, the CCR2/5 inhibition, using a novel oral CCR2/5 antagonist (BMS-687681), strongly reduced the number of MoMF, and led to improved liver fibrosis. Results from the second year of follow-up in the CENTAUR trial suggested that the antifibrotic effect of cenicriviroc might not be durable in the long-term [19], possibly because the underlying metabolic stress is not alleviated. One could speculate that the combination of CCR2/5 antagonists with a metabolic, antisteatotic drug (such as an FGF21 agonist) can improve NASH and fibrosis in human patients more effectively than either drug alone [34], as we have shown in this study using a fibrotic NASH mouse model.

FGF21 belongs to the endocrine and paracrine subfamily of FGFs that also include FGF15, 19 and 23. FGF21 is synthesized in the liver and released into the systemic circulation. The co-receptor β-Klotho is essential for FGF21 activity and downstream effects [42]. FGF21 gains organ specificity, as the co-receptor is mainly produced in the liver and white adipose tissue, so that FGF21 stimulates glucose uptake in adipocytes and lowers triglyceride levels in rodents [21,22]. In line with our findings in the CDAHFD mouse model, previous studies showed that FGF21 successfully amended obesity and diabetes in a high-fat diet (HFD) model [24]. Ongoing clinical trials demonstrated significantly reduced content of hepatic triglycerides in NASH patients treated with FGF21 agonists compared to the placebo group [27]. The efficacy and safety of the FGF21 analogue pegbelfermin has been evaluated in two phase 2b clinical study in patients with NASH and stage 3 fibrosis (FALCON1, ClinicalTrials.gov Identifier NCT03486899) and patients with compensated NASH cirrhosis (FALCON2, ClinicalTrials.gov Identifier NCT03486912) [28]. Both studies could not reach primary endpoints (≥1 stage improvement in fibrosis without NASH worsening after 24 weeks or 48 weeks)—however, pegbelfermin treatment resulted in higher rates of fibrosis (≥1 stage reduction in 27% of the patients dosed at 40 mg SQ q.wk.) and NASH improvement (hepatic fat content (HFF) ≥10% reduction in 23% of the patients dosed at 40 mg SQ q.wk.). In addition, pegbelfermin administration showed beneficial effects in both studies based on various non-invasive surrogate markers (decrease of liver transaminases and plasma pro-peptide of type lll collagen, increase of adiponectin concentrations) [29,30]. However, further development of pegbelfermin was terminated for non-cirrhotic NASH, as no clear dose-dependent reduction in liver fibrosis could be demonstrated by the single agent regimen.

A currently open question in the field is whether treatment strategies against NAFLD should be personalized based on gender/sex. Male individuals predominantly show more severe stages of NAFLD such as NASH and fibrosis than female individuals during the reproductive age. However, after menopause, NAFLD occurs at a higher rate in women, supporting that estrogen is protective [43]. In the gender specific analyses we observed a significant correlation in male NAFLD patients between CCL2 serum levels and advanced fibrosis stages as well as a positive trend for female patients (*p* = 0.064). Of note, CCL2 serum concentrations did not correlate with biopsy proven NASH activity (NAS). FGF21 levels were associated with biomarkers of NASH and histopathologic scoring (F3-F4 fibrosis) in female patients but we did not observe a significant correlation based on the whole cohort. Nonetheless, we have to be careful interpreting the data based on our patient cohort, because the above-mentioned gender discrepancy was also apparent in our clinical cohort resulting in a low statistical power especially in the female population. As expected from the literature, our cohort that was enriched for advanced disease stages displayed a lower number of female than male patients (total female patients *n* = 20 vs. *n* = 65 males).

In this project, we employed the CDAHFD model to induce NAFLD [32]. While the ideal NAFLD model does not exist, there is consensus that the experimental model should reflect key characteristics of human disease [10]. CDAHFD fed mice develop steatohepatitis and severe liver fibrosis over a relatively short period of time. In contrast to the more commonly used methionine-choline deficient (MCD) dietary model, mice do not experience a drastic weight loss that is typical for the MCD diet, but recover to baseline levels after an initial weight loss [44]. However, the therapeutic effects of PEG-FGF21v on steatosis and metabolism can presumably not be sufficiently studied in MCD diet. On the other hand, effects on fibrosis improvement by CCR2/CCR5 inhibition have been reported before in the MCD diet model [17], to a similar extent as we describe it now for the CDAHFD. Additionally, the PEG-FGF21v as monotherapy has been demonstrated to show therapeutic benefit on parameters of weight loss, steatosis and fibrosis in the CDAHFD mouse [45]. However, the CDAHFD model lacks some features of the metabolic syndrome such as obesity or insulin resistance. Thus, further studies should aim at addressing effects of combination therapy on “extrahepatic metabolic diseases” including hyperinsulinemia, atherosclerosis or cardiovascular diseases.

In conclusion, this study confirms the therapeutic efficacy of CCR2/5 antagonists and FGF21 agonists in an experimental model of steatohepatitis and fibrosis. Additionally, we demonstrated that targeting inflammatory and metabolic pathways at the same time ameliorated various aspects of NAFLD to a greater extent than single drug treatment alone. Our data suggest that combination therapies bear the potential of additive effects in the course of disease progression, supporting the hypothesis of multiple pathophysiologic triggers that need to be addressed in parallel in the treatment of NAFLD. Therefore, further studies seem warranted to test combinations of different drug targets in human NAFLD and NASH.

## 4. Materials and Methods

### 4.1. Patient Cohort

Patients with biopsy-proven NAFLD (*n* = 85) were prospectively recruited at the Ghent University Hospital, Belgium, between 2011 and 2018, as previously described [46]. Appropriate exclusion of liver disease of other etiologies, including alcohol-induced or drug-induced liver disease, viral or auto-immune hepatitis, metabolic and cholestatic liver diseases, was performed using specific clinical, biochemical, histological and/or radiographic criteria. All patients were caucasian and had a negative history of alcohol abuse as indicated by an average daily alcohol consumption of ≤20 g. None of the subjects were on treatment with corticosteroids or insulin. After applying these exclusion criteria, we included 85 patients for analysis.

Blood samples were collected after overnight fasting. All samples were centrifuged, fractionated and serum stored at −80 °C until further analysis. Laboratory evaluation included standard liver biochemistry (alanine aminotransferase (ALT), aspartate aminotransferase (AST), γ-glutamyl transpeptidase (GGT)), complete blood count, triglycerides and total serum cholesterol. Body mass index (BMI) was calculated as body weight/height^2^ (kg/m^2^). Diabetes mellitus was defined according to the American Diabetes Association criteria [47].

Human liver biopsies were routinely processed and stained with hematoxylin-eosin (H&E) and Sirius red. An experienced pathologist (A.H.) evaluated the biopsies, blinded to the patient characteristics. Only biopsies with at least 6 complete portal tracts were deemed appropriate for adequate histological evaluation. Histological features were scored according to the NASH Clinical Research Network scoring system [48]. A diagnosis of non-alcoholic fatty liver (NAFL) was made if ≥5% of hepatocytes contained macrovesicular lipid droplets, whereas the diagnosis of NASH was based on the joint presence of steatosis, hepatocyte ballooning and lobular inflammation [49,50]. Fibrosis was evaluated using the NASH Clinical Research Network fibrosis staging system [48].

The study protocol was approved by the Ghent University Hospital Ethical Committee and conducted according to the principles of the Declaration of Helsinki. Participants gave their written informed consent, which was validated by the Ethical Review Board.

### 4.2. Animal Experiments

7-week-old C57BL6/J wildtype mice (Janvier Labs, Le Genest-Saint-Isle, France) were housed in a specific-pathogen-free environment at the Animal Facility of the University Hospital Aachen in a 12-h light/dark cycle with free access to food and water. In vivo animal experiments were performed with male mice at eight weeks of age under conditions approved by the appropriate institutional and governmental authorities according to German legal requirements (State Agency for Nature, Environment and Consumer Protection in North-Rhine Westphalia, LANUV NRW).

### 4.3. Pharmacological Treatment and Induction of Liver Injury

Both pharmacologic compounds were kindly provided by Bristol-Myers-Squibb. The CCR2/5 antagonist (BMS-687681) was dissolved in sterile water at pH 3 containing 0.5% methylcellulose (400 cps) and 0.1% Tween-80. The CCR2/5 antagonist was administered via oral gavage (PO) at either 45 mg/kg body weight (BW) b.i.d. in single drug treatment or 15 mg/kg BW b.i.d. in combination treatment. PEG-FGF21 variant (BMS-986171) was suspended in a vehicle containing 20 mM Tris(hydroxymethyl)aminomethane and 250 mM sucrose at pH 8.3. PEG-FGF21v was administered by subcutaneous (SC) injection at 0.6 mg/kg BW twice weekly.

Carbon tetrachloride (CCl_4_) (Merck, Darmstadt, Germany) solved in corn oil was injected once intraperitoneally (IP) at 0.6 mL/kg BW to induced acute liver injury. All mice were sacrificed after 36 h and liver and blood samples were retrieved for analysis.

As a representative NAFLD model, mice were fed a choline-deficient, L-amino acid-defined, high-fat diet (CDAHFD) (A06071302, Research Diets, New Brunswick, NJ 08901, USA) for up to 12 weeks. Pharmacologic treatment started as single drug or combination therapy after 6 weeks of diet administration. Mice were sacrificed after two or six weeks of treatment for final analysis.

### 4.4. Phenotypic Assessment and Model Endpoints

Conventional hematoxylin-eosin (H&E) and sirius red stainings were performed according to established protocols and necrotic area fraction was quantified with ImageJ [16]. NAFLD activity score (NAS) was assessed by a medically qualified investigator blinded to the treatment groups, and colometric tests were conducted for hydroxyproline. Immunohistochemistry stainings for F4/80 (Abcam) were performed on paraffin-embedded liver sections [51]. Alanine aminotransferase (ALT) and aspartate aminotransferase (AST) activities were measured (UV test at 37 °C) in serum (Roche Modular pre-analytics system, Rotkreuz, Switzerland).

### 4.5. Flow Cytometry of Mouse Samples

Liver and blood leukocytes were analyzed by multicolor flow cytometry using an LSR-Fortessa (BD Biosciences), as described [16]. Livers were perfused with cold phosphate buffered saline (PBS) (Pan Biotech, Aidenbach, Germany), homogenized and digested by collagenase type IV (Worthington Biochemical Corporation, Lakewood, NJ, USA) in a heated bath (37 °C). Leukocytes were then isolated by Nycodenz gradient (Alere technologies, Oslo, Norway) differential centrifugation steps. The cells were stained with fluorochrome-conjugated antibodies, employing a myeloid (CD31/FITC, CD4/FITC, CD3/FITC, CD19/FITC, Ly6G/FITC, CD68/PE, CD11b/PERCP-Cy5.5, F4/80/PE-Cy7, Tim4/APC, Gr1/APC-Cy7, MHCII/V450, CD45/V500, CX_3_CR1/BV711 and 7-AAD/PE-Cy5) and a lymphoid (CD31/FITC, Ly6G/FITC, CD19/PERCP-Cy5.5, TCRβ/PE-Cy7, CD44/APC, NK1.1/APC-Cy7, CD4/V450, CD45/V500, CD8a/BV711 and 7-AAD/PE-Cy5) panel (Appendix A).

Whole blood was subjected to red cell lysis by Pharmlyse (BD Biosciences, San Jose, CA, USA) and stained with a mixed myeloid-lymphoid panel (Ly6G/FITC, Gr1/PERCP-Cy5.5, CD115/PE, TCRβ/PE-Cy7, CD11b/APC, NK1.1/APC-Cy7, CD19/Al700, CD4/V450, and CD8a/BV711). After staining, the samples were analyzed with the LSR Fortessa (BD Biosciences) and FlowJo v10.2 (FlowJo LLC, BD Biosciences, Ashland, Oregon 97520, USA) (Appendix A). Counting beads (BD Biosciences) were added to single-cell suspensions to determine absolute cell numbers in liver and blood.

### 4.6. Multiplex Magnetic Bead Assay

Mouse chemokine (C-C motif) ligand (CCL) 2, CCL5, CXCL1 and interleukin (IL)-10 serum protein levels were determined with a multiplex bead-based assay (Bio-Plex^®^ MAGPIX™ Multiplex Reader, Bio-Rad, Temse, Belgium), using coupled beads from mouse cytokine group I (Bio-Rad). The reported assay sensitivity, intra- and inter-assay coefficients of variation are 3.7 pg/mL, 5% and 7% for CCL2, 0.6 pg/mL, 4% and 4% for CCL5, 0.3 pg/mL, 3% and 30%, for CXCL1, and 1.0 pg/mL, 4% and 5% for IL-10, respectively.

### 4.7. RNA Extraction and Quantitative Real-Time qPCR

RNA was extracted from 20 mg mouse using the RNeasy plus mini kit (Qiagen), according to the manufacturer’s protocol. The RNA quality was evaluated by spectrophotometry (Nanodrop, Thermo Fisher Scientific, Ghent, Belgium), calculating the A260/A280 ratio. cDNA synthesis was performed starting from 1 µg RNA, using the SensiFAST cDNA synthesis kit (Bioline, London, UK). cDNA was added to a 384-well plate with specific primers (Biolegio, Nijmegen, The Netherlands) (Appendix A) and Sensimix SYBR No-ROX Mastermix (Bioline). Samples were run and analyzed on the Lightcycler 480 II (Roche). PCR reactions using water instead of template showed no amplification. Measurements were performed in duplicate and Cq values were calculated with the second derivative maximum method. Average Cq values were normalized to the Cq of stable housekeeping genes, according to analysis in GeNorm (Biogazelle, Ghent, Belgium).

### 4.8. Enzyme-Linked Immunosorbent Assay

Serum CCL2 and FGF21 concentrations in human serum were determined using Human Quantikine ELISA kits (DCP00 and DF2100 respectively, R&D, Oxon, UK) according to the manufacturer’s protocols. Human cytokeratin-18 M30 fragments were measured using the M30 Apoptosense ELISA kit (TECOmedical, Nijkerk, The Netherlands).

### 4.9. Statistics

Statistical analysis was performed using SPSS 25.0 (SPSS Software, IBM Corp., Armonk, NY, USA) and GraphPad Prism 6 (GraphPad Software Inc., La Jolla, CA, USA). The appropriate parametric or non-parametric tests were applied. A two-tailed *p* value < 0.05 was considered statistically significant. Continuous variables are presented as median (interquartile range) or mean ± standard deviation (SD), depending on the normality of distribution.

For analysis of patient data, significant correlations were determined by calculating the Spearman’s correlation coefficient. Multivariate binary logistic regression analysis on variables significantly associated with advanced fibrosis in univariate analysis, followed by stepwise backward elimination, was performed to identify factors independently associated with the presence of advanced fibrosis.

All experimental data from mice are presented as mean ± SD. Differences between groups were evaluated by two-tailed unpaired Student *t*-test, one-way ANOVA and Pearson’s linear correlation analysis (GraphPad Prism 6).

## Figures and Tables

**Figure 1 ijms-23-06696-f001:**
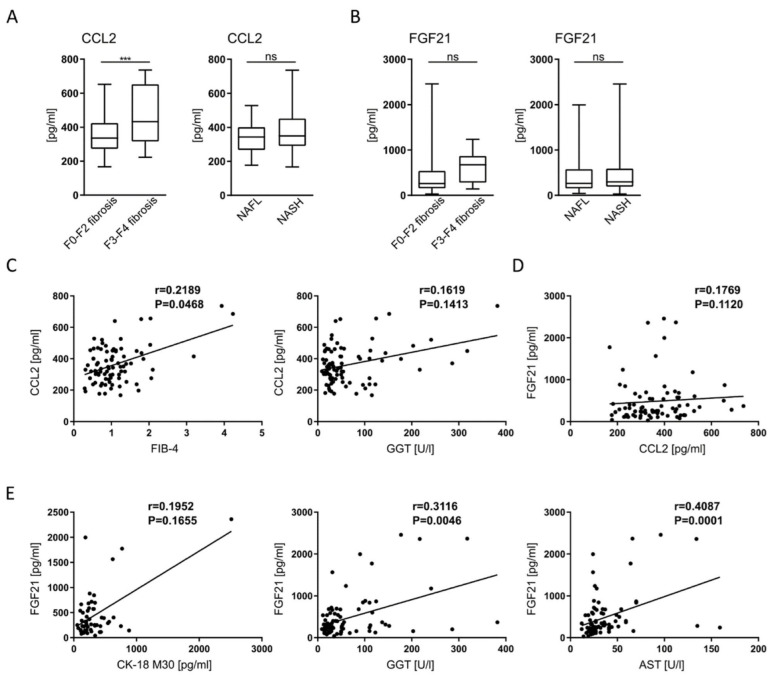
Correlation of CCL2 and FGF21 serum levels with severity of human NAFLD. Serum samples were obtained from patients with biopsy-proven NAFLD (*n* = 85). (**A**,**B**) CCL2 and FGF-21 serum levels measured by ELISA and correlated with histologically assessed severity of liver fibrosis and steatohepatitis. (**C**,**D**) Correlation of CCL2 serum concentrations with biomarkers of fibrosis (FIB-4 and GGT) and FGF-21 serum levels. (**E**) Association of FGF-21 serum levels with biomarkers of steatohepatitis (CK-18 fragment M30, GGT and AST). ns = non-significant, *** *p* < 0.001 (unpaired Student *t* test in (**A**,**B**) Spearman’s r and *p*-values of linear correlation analysis in (**C**–**E**)).

**Figure 2 ijms-23-06696-f002:**
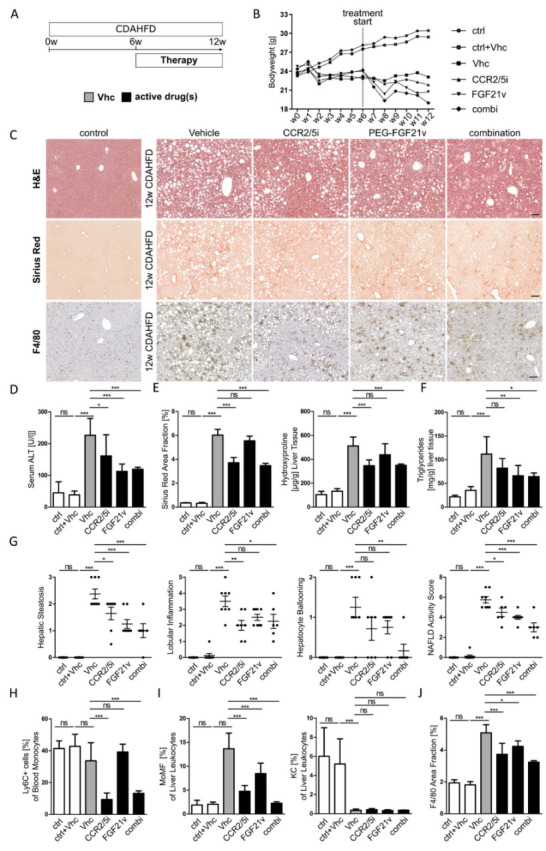
Combination therapy by dual CCR2/CCR5 inhibition and FGF21 agonism ameliorates steatohepatitis and fibrosis more effectively than single drug treatment. (**A**) Pharmacologic treatment with CCR2/CCR5 inhibitor (CCR2/5i) and/or PEG-FGF21 variant (FGF21v) was conducted over the last 6 weeks of 12 weeks CDAHFD (choline-deficient, amino acid-defined high-fat diet) administration to induce steatohepatitis and fibrosis. (**B**) Line graph of bodyweight development of all treatment groups (ctrl: control diet; Vhc: vehicle). (**C**) Representative H&E, Sirius Red and F4/80 immunohistochemistry staining (×10 magnification; scale bars = 100 µm. (**D**–**F**) Assessment of liver injury by serum alanine transaminase levels (ALT), of liver fibrosis by quantification of Sirius Red area fraction and hydroxyproline content and of hepatic triglyceride content. (**G**) Single parameters of the histopathological NAFLD activity score (NAS). (**H**–**J**) Quantification of flow cytometry for Ly6C+ blood monocytes, hepatic monocyte-derived macrophages (MoMF) and Kupffer cells (KC) and quantification of F4/80 positive area fraction. All data are presented as mean SD (*n* ≥ 6 per group), ns = non-significant, * *p* < 0.05, ** *p* < 0.01, *** *p* < 0.001 (one-way ANOVA with post-hoc testing).

**Figure 3 ijms-23-06696-f003:**
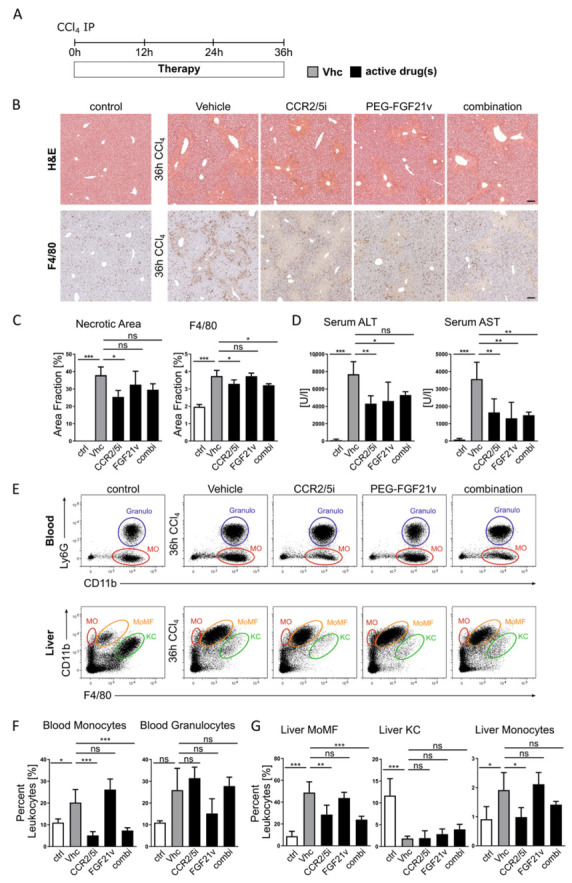
Blocking hepatic infiltration of monocytes and macrophages by CCR2/CCR5 inhibition in acute liver injury. (**A**) Acute liver injury was induced by a single CCl_4_ injection. Mice (*n* = 4 per group) received vehicle (Vhc), CCR2/CCR5 inhibitor (CCR2/5i) and/or PEG-FGF21 variant (FGF21v). Liver injury and immune cell migration was assessed 36 h after injury induction. (**B**) H&E and F4/80 immunohistochemistry staining of representative liver sections of control and treatments groups (×10 magnification; scale bars = 100 µm). (**C**,**D**) Quantification of F4/80 positive area fraction. Hepatic injury was assessed by necrotic area fraction and serum alanine (ALT) and aspartate (AST) transaminase levels. (**E**–**G**) Representative flow cytometric plots of blood (MO = monocytes; Granulo = granulocytes) and liver (MO = monocytes; MoMF = monocyte-derived macrophages; KC = Kupffer cells) immune cell populations and corresponding quantification. Data are presented as mean ± SD (*n* = 6–8 per group), ns = non-significant, * *p* < 0.05, ** *p* < 0.01, *** *p* < 0.001 (one-way ANOVA with post-hoc testing).

**Figure 4 ijms-23-06696-f004:**
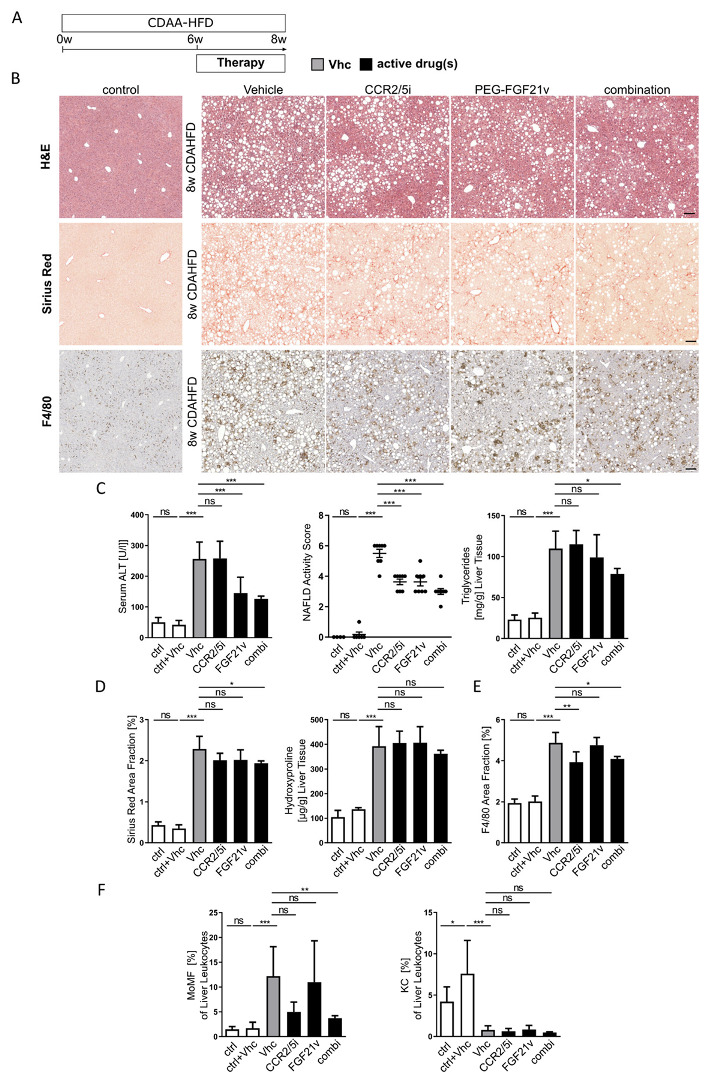
Beneficial effects of combination therapy after short term treatment. (**A**) Steatohepatitis and fibrosis were induced by CDAHFD (choline-deficient, amino acid-defined high-fat diet) over a total period of 8 weeks. Effects of pharmacologic treatment were assessed after administration of CCR2/CCR5 inhibitor (CCR2/5i) and/or PEG-FGF21 variant (FGF21v) over the last two weeks of injury induction. (**B**) Representative liver section of H&E, Sirius Red and F4/80 immunohistochemistry staining (×10 magnification; scale bars = 100 µm). (**C**,**D**) Serum alanine transaminase (ALT) levels, NAFLD activity score, hepatic triglyceride and hydroxyproline content as well as quantification of Sirius Red area fraction display the liver phenotype. (**E**,**F**) Quantification of F4/80 positive area fraction and flow cytometrically determined monocyte-derived macrophages (MoMFs) and liver Kupffer cells (KC). All data are presented as mean SD (*n* ≥ 6 per group) ns = non-significant, * *p* < 0.05, ** *p* < 0.01, *** *p* < 0.001 (one-way ANOVA with post-hoc testing).

**Table 1 ijms-23-06696-t001:** Patient characteristics.

Characteristic	NAFL (*n* = 31)	NASH (*n* = 54)
Age, years	44 (30–49)	49 (40–59)
Sex (male/female)	24/7	41/13
BMI, kg/m^2^	41.5 (39.4–45.6)	38.5 (34.9–42.8)
Type 2 diabetes, presence	6 (19.4)	30 (55.5)
Triglycerides, mg/dL	170 (124–199)	185 (142–257)
Total Cholesterol, mg/dL	182 (157–207)	169 (137–220)
Thrombocytes, ×10^3^/µL	241 (210–274)	219 (195–267)
AST, U/L	24 (22–30)	33 (25–51)
ALT, U/L	36 (30–49)	46 (32–75)
GGT, U/L	29 (19–45)	50 (28–110)
CCL2, pg/mL	342.8 (267.0–400.3)	349.5 (291.0–451.8)
FGF-21, pg/mL	265.6 (156.4–573.1)	296.4 (189.8–587.3)
CK-18 M30, U/L	252.7 (184.0–340.4)	270.9 (177.7–480.0)

Results are expressed as mean ± SD or median (interquartile range) for continuous variables, depending on the normality of the distribution, and *n* (%) for categorical variables. ALT: Alanine Aminotransferase; AST: Aspartate Aminotransferase; BMI: body mass index; CCL2: chemokine (C-C motif) ligand 2; CK-18 M30: cytokeratin-18 M30 fragments; FGF-2: fibroblast growth factor 21; GGT: γ-glutamyltransferase.

## Data Availability

Not applicable.

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
