# Peer review of "Combined Therapy with a CCR2/CCR5 Antagonist and FGF21 Analogue Synergizes in Ameliorating Steatohepatitis and Fibrosis"

_ijms, 2022, doi:10.3390/ijms23126696_

Round 1

Reviewer 1 Report

The paper by Puengel and colleagues concerns the use of CCR2 / CCR5 antagonists and FGF21 analogues in the treatment of non-aloholic fatty liver disease (NAFLD) and non-alcoholic steatohepatitis (NASH). NAFLD and NASH are pathologies of great hepatological interest because, not only its incidence is constantly growing, but also because in 20/30% of cases, NASH can lead to the development of hepatocellular carcinoma or, more rarely, cholangiocarcinoma. Furthermore, currently available therapies for the treatment of NASH are dismal and it is impossible to reduce fibrotic scar, especially in the advanced stages of disease. The paper is clear and well written and the data is very interesting. Furthermore, the results are convincing and support the idea behind the work. The main results obtained by Authors are: 1. Serum concentrations of CCL2 and FGF21 correlate with different disease severity readouts. 2. Using NASH models, in vivo treatments with CCR2/5 inhibitors (CCL2 receptors), with FGF21 analogues or with a mix of them, induce a general remission of the disease and in particular of MoMF recruitment. 3. Similarly to what was seen in the mouse model of NASH (chronic damage), even in case of acute damage with CCl4, the treatment with the same compounds gives a relief of the disease.

I have a few concerns about this job:

·       Since NAFLD/NASH show clear gender differences, with a higher incidence on the male population and since the Authors used only male mice for the in vivo experiments, it would be interesting to evaluate the data of figure 1 (CCL1 and FGF21 secretion and correlation with disease readouts), not only as a single group, but also splitted by gender.

·       To confirm the data in supplementary figures 1 and 2B, since an IHC was done for macrophages (f4-80), I would ask to do IHC for other immune cell populations on liver sections of control and treated mice.

·       A graphical abstract or an explanatory drawing would be very much appreciated, in order to better clarify the mechanism that has been demonstrated.

Reviewer 2 Report

To explore the pathogenesis and treatment of NASH are liver diseases research hotspot. NASH are charactered by steatosis and hepatitis, and some cases could develop to fibrosis/cirrhosis and HCC. In this article, CDAHFD was used to set up mice NASH plus fibrosis model. Authors set up the combined therapy with CCR2/CCR5 antagonist (BMS-687681) and FGF21 analogue synergizes PEG-FGF21 variant (BMS-986171) to inhibit inflammatory reaction and reduce lipid deposition in liver. HE, IHC of F4/80 and Sirius Red were used to observe liver pathological changes. FACS was used to check inflammatory cells changes in peripheral blood and liver. Combined therapy ameliorated steatohepatitis and fibrosis, and they are new approaches to treat NASH patients. It is quite an interesting research article and would help us explore the innovative approach to treat the increasing NAFLD and NASH; however, I still have a few concerns regarding the manuscript.

      1.       For the abbreviated word appearing in the 1st time, the whole name should be shown. For example, the whole name of CDAHFD should be shown in Abstract or Introduction, not in the legend of Fig 2.

2.       Fig 1 A, human serum CCl2 concentration positively correlated with liver fibrosis level. Whether did liver inflammatory level correlate with serum CCl2?

3.       In Materials and Methods, 4.5. Flow cytometry of mouse samples. All used antibodies and their fluorescence color have been listed, but please list the positive and negative marker of each cell, such as KC would be F4/80HighLy6CLowCD45+   

Reviewer 3 Report

The study by Puengel et al. describes a two-pronged approach for alleviating NASH and aspects of acute liver injury by targeting CCR2/5 and FGF21.  The data includes human patient data with NAFLD/NASH at different fibrotic stages, multiple mouse models using the CDAHFD diet, and an acute injury CCl4 model.  Overall, the study is well designed in describing how combined therapy may be useful for targeting metabolic (FGF21) and inflammatory (CCR2/5) components of inflammatory fatty liver disease.  The addition of characterization of blood and liver immune cells adds a mechanistic approach which is a strength of this study.  That being said, there are areas the authors need to address to improve their study:

1)      My biggest concern with the data as presented has to do with the CDAHFD model and the conclusions drawn from it.  Did all of the CDAHFD groups have similar consumption of the diet?  All of the treatments had a weight drop after the treatment started in week 6.  In particular the combined group went from ~24g to ~19g from weeks 6-8.  This supports that they likely could have had reduced food intake following treatment.  Without reporting this data, the conclusion could be made that all effects in the CDAHFD model were due to changes in food consumption, which reduced their exposure to the CDAHFD diet, and therefore showed anti-inflammatory and reduced hepatic lipid deposition effects compared to the vehicle group.  Food consumption must be reported to alleviate this potential alternative hypothesis.

2)      For human data, it would have strengthened this report to have BMI/sex-matched control patients and report their CCL2, FGF21 and CK18 values.

3)      Validation of CCL2/5 inhibition should be shown by showing a reduction of downstream signaling such as MAPK.  The other possibility would be to show that CCL2 and CCL5 mRNA were unchanged in liver following treatment with inhibitors and then subsequently adding this data to supplemental figure 1E.

4)      Was there any mortality in the CCR2/5i + FGF21 groups?  Their weights at 12 weeks appear to be dropping?  Was this weight drop significant compared to eh CDAHFD Vhc group?  Was there any change in appearance or overall well being of the mice?  This needs to be reported in the text.

5)      With the protective effects of FGF21v for some measures in the CCl4 model, is it possible that FGF21v treatment changes metabolism of CCl4 by changing cytochrome p450 activity (such as CYP2E1)?  This needs to be checked for both CCR2/5i and FGF21v groups to ensure conclusions of these groups are appropriate.

Round 2

Reviewer 1 Report

The Authors satisfactorily answered the question raised by the reviewer. The paper is definitively improved and worthy of publication.

Reviewer 3 Report

The authors have answered my primary concern with this study which was the use of the CDAHFD diet and potential complications of mice.  No model is perfect and the fact there was no mortality associated with the weight drop in some treatment groups alleviates my major concern regarding the potential of combined therapy for patient populations in the future.  While I would have preferred some measures to be included (such as food consumption) the authors adequately explained rationale for not being able to do so.  I have no significant remaining concerns with this study.